# Tough and Robust Metallosupramolecular Hydrogels Enabled by Ti_3_C_2_T_x_ MXene Nanosheets

**DOI:** 10.3390/polym15194025

**Published:** 2023-10-09

**Authors:** Biqiang Jin, Wenqiang Wu, Zhaoyang Yuan, Changcheng Wang

**Affiliations:** 1College of Science, Xichang University, Xichang 615000, China; 2State Key Laboratory of Polymer Materials Engineering, College of Polymer Science and Engineering, Sichuan University, Chengdu 610065, China; zhaoyangyuan@stu.scu.edu.cn (Z.Y.); wcc.x@foxmail.com (C.W.); 3Sichuan Dowhon New Material Co., Ltd., Chengdu 610036, China

**Keywords:** Ti_3_C_2_T_x_ MXene, coordination interaction, physical cross-links, metallosupramolecular hydrogels

## Abstract

Recently, many tough synthetic hydrogels have been created as promising candidates in fields such as smart electronic devices. In this paper, we propose a simple strategy to construct tough and robust hydrogels. Two-dimensional Ti_3_C_2_T_x_ MXene nanosheets and metal ions were introduced into poly(acrylamide-co-acrylic acid) hydrogels, the MXene nanosheets acted as multifunctional cross-linkers and effective stress-transfer centers, and physical cross-links were formed between Fe^3+^ and carboxylic acid. Under deformation, the coordination interactions exhibit reversible dissociation and reorganization properties, suggesting a novel mechanism of energy dissipation and stress redistribution. The design enabled the hydrogel to exhibit outstanding and balanced mechanical properties (tensile strength of up to 5.67 MPa and elongation at break of up to 508%). This study will facilitate the diverse applications of metallosupramolecular hydrogels.

## 1. Introduction

Hydrogel, a polymer network using water as a solvent, has shown great potential for application in various emerging intelligent devices and other fields, and it has received much attention from researchers in recent years [1,2]. However, the low strength and poor toughness of traditionally developed hydrogels, caused by covalent cross-linking and an inhomogeneous network, severely limit their applications in areas such as load bearing [3]. The development of novel hydrogels with comprehensive properties similar to those of natural load-bearing materials is desirable. This can be achieved through different network topologies and energy dissipation mechanisms. Tremendous efforts have been made in this regard, such as double-network hydrogels [4,5], slip-ring hydrogels [6,7], macromolecular cross-linked hydrogels [8], microgel-reinforced hydrogels [9,10], and supramolecular hydrogels [11,12,13,14,15,16,17,18,19].

Among these robust hydrogels, metallosupramolecular hydrogels have emerged as a promising class of materials due to their unique combination of mechanical strength, self-healing ability, and responsiveness to external stimuli. These hydrogels are formed through the coordination of metal ions with supramolecular coordination bonds, resulting in the formation of a three-dimensional network structure. For example, Zhou et al. [20] prepared a series of robust and tough hydrogels by utilizing the coordination of carboxylate and Fe^3+^. In the presence of few chemical cross-linkers, acrylamide and acrylic acid aqueous solutions were subjected to free-radical polymerization to prepare poly(acrylamide-co-acryl acid) hydrogels that were only chemically cross-linked, which were subsequently immersed in an Fe^3+^ solution to form carboxylate-Fe^3+^ coordination bonds for secondary cross-linking. Finally, the second cross-linked hydrogels were transferred into a large amount of deionized water to remove the Fe^3+^ ions not involved in the coordination cross-linking and to form more tridentate coordination to obtain the carboxylic acid-Fe^3+^ double-cross-linked hydrogels with the best performance. Mechanical property tests showed that the mechanical properties of the hydrogels were not strongly related to the first chemical cross-linking but closely related to the content of acrylic acid and the concentration of Fe^3+^. Before the introduction of coordination bonds into the hydrogel, the tensile strength was only 100 kPa, and the loading–unloading cyclic curve showed elastic characteristics with almost no hysteresis, i.e., there was almost no energy dissipation in the chemically cross-linked hydrogel at this time; and after introducing Fe^3+^, coordination bonds were formed in the hydrogel network, its tensile strength was as high as 6 MPa, and its toughness was as high as 27 MJ/m^3^. Wu et al. [21] developed a high-strength and high-toughness poly(acrylamide-co-acrylic acid) hydrogel based on the coordination cross-linking between carboxylate radicals and Fe^3+^ by the solution casting method. The hydrogel had a water content of 35–85%, and its fracture strength, fracture strain, Young’s modulus, and fracture toughness were as high as 18 MPa, 1100%, 80 MPa, and 40 MJ/m^3^, respectively. By adjusting the coordination process FeCl_3_ solution’s pH, concentration, and content of carboxylic acid units, the mechanical properties of hydrogels could be effectively regulated. Due to the carboxylic acid-Fe^3+^ coordination bond dynamics, the hydrogel exhibited obvious tensile rate dependence, temperature dependence, and a good self-recovery ability. Meanwhile, cyclic tensile tests were performed on this carboxylate/iron ion coordination cross-linked hydrogel. The large hysteresis circles surrounding the stress–strain curves and the obvious hysteresis behavior proved that a strong energy dissipation mechanism was introduced into the hydrogel network. Recently, Wu et al. [22] designed and fabricated a class of physical hydrogels based on coordination cross-linking between poly(acrylamide-maleic acid) and Fe^3+^. The physical hydrogels showed linear viscoelasticity over the strain range of 0–1700%, with tensile strengths of up to 12.0 MPa and toughness of up to 82.1 MJ/m^3^. A key design principle in the preparation of the hydrogels was the introduction of a unique monomer, maleic acid, copolymerized with acrylamide. The maleic acid units are uniformly distributed along the backbone of the acrylamide polymer chain. This is essential for the formation of a homogeneous network after the introduction of carboxylate-Fe^3+^ coordination bonds. Immediately following this work, Wu et al. [10] further introduced carboxyl-rich spherical nanogels into the hydrogel network to construct a special nano-domain reconfiguration topologic network. Interpenetrating linear carboxylated polyacrylamide chains in poly(N-isopropylacrylamide-co-acrylic acid) nanogels are cross-linked by coordination bonds to form the network. At small strains, the interpenetrating polymer chains are capable of sticky slipping motions; at large strains, the coordination interactions and nanogels experience fracture and reorganization. This special design efficiently dissipates energy and redistributes stress concentrations, resulting in hydrogels with excellent mechanical properties: a fracture stress of up to 10.8 MPa, fracture strains of up to 2048%, and toughness of up to 111.8 MJ/m^3^.

In recent years, there has been a growing interest in developing metallosupramolecular hydrogels with enhanced toughness and robustness to overcome the limitations of conventional hydrogels [7,23,24,25]. One approach to achieve this is by incorporating nanosheets into the hydrogel matrix, which can provide additional mechanical reinforcement and improve the overall performance of the hydrogel. MXene nanosheets, a new type of two-dimensional material, have shown great potential in various applications due to their exceptional mechanical properties, high electrical conductivity, and good chemical stability [26]. In particular, Ti_3_C_2_Tx MXene nanosheets have attracted significant attention for their ability to enhance the mechanical properties of hydrogels. For instance, Xie et al. [1] prepared ultra-tough and highly stretchable nanocomposite physical hydrogels based on the multibond network design rationale, in which two-dimensional Ti_3_C_2_T_x_ MXene nanosheets acted as multifunctional cross-linkers and effective stress transfer centers. The prepared MXene-polyacrylic acid-Fe^3+^ physical hydrogels exhibited outstanding mechanical properties: tensile strength of up to 10.4 MPa and elongation at break of up to 3080%. This is attributed to the dual cross-linking network: dense Fe^3+^-mediated coordination cross-links between MXene nanosheets and polyacrylic acid chains, and sparse carboxyl-Fe^3+^ cross-links between polyacrylic acid chains.

Herein, tough and robust metallosupramolecular hydrogels were enabled by Ti_3_C_2_T_x_ MXene nanosheets. Through in situ free-radical polymerization, two-dimensional MXene nanosheets were introduced as multifunctional cross-linkers to strengthen the mechanical properties of the obtained hydrogels. Fe^3+^ was introduced to form coordination cross-links, which substantially dissipate mechanical energy and redistribute stress concentration. The design of the dual-cross-linked network incorporates both Fe^3+^-mediated coordination cross-links between the MXene nanosheets and the carboxyl-rich polyacrylamide chains, and carboxyl-Fe^3+^ coordination cross-links between the polymer backbone chains. This strategy resulted in excellent mechanical properties of the prepared metallosupramolecular hydrogels. Moreover, due to the good electrical conductivity of MXene, the hydrogels reported in this paper have a broad application potential in flexible robots and wearable devices.

## 2. Materials and Methods

### 2.1. Materials

Acrylamide (AM) and ferric (III) chloride hexahydrate (FeCl_3_·6H_2_O) were purchased from Chengdu Huaxia Chemical Reagent Co., Ltd., Chengdu, China. Acrylic acid (AA), Ti_3_AlC_2_ (MAX), lithium fluoride (LiF), and concentrated hydrochloric acid (HCl), potassium persulfate (KPS), and tetramethylethylenediamine (TMEDA) were obtained from Adamas, Shanghai, China. Deionized water was homemade in the laboratory.

### 2.2. Synthesis of MXene (Ti_3_C_2_T_x_) Nanosheets

First, 1 g of LiF was mixed with 20 mL of 9 mol L^−1^ HCl solution. Next, 1 g of Ti_3_AlC_2_ (MAX) was slowly added to the above mixed solution, accompanied by stirring to avoid intense exotherm. Subsequently, the solution was washed with deionized water and centrifuged for 5 min until the pH of the supernatant was higher than 5. Finally, the collected precipitate was sonicated for 30 min and centrifuged for 1 h to obtain the MXene nanosheets suspension [1].

### 2.3. Synthesis of Hydrogels

The hydrogels were synthesized by in situ free-radical polymerization. In brief, 30 g of AM, 4.56 g of AA, different mass fractions of MXene aqueous solution (0.2 wt%, 0.4 wt%, 0.6 wt%, 0.8 wt%), and 25 μL TMEDA were dissolved in deionized water. The total concentration of all reaction monomers in the precursor solution was fixed at 4.5 mol/L. Nitrogen was vented for 10 min until the dissolved oxygen was completely removed from the precursor solution, then 60 mg of KPS was added to the reaction solution as an initiator. The solution was then transferred to a water bath and stored at 25 °C for 10 h without any outside interference whatsoever. Next, the samples were immersed in 0.1 mol/L FeCl_3_ solution for 10 h at room temperature to form the original Fe^3+^-carboxylic acid cross-links, i.e., coordination process. Finally, the above premier hydrogels were soaked in excess deionized water for 24 h (with fresh deionized water changes at 3 h intervals) to remove excess Fe^3+^ ions and form more tridentate ligands to obtain the optimal tough and robust metallosupramolecular hydrogels.

### 2.4. General Characterization

The structure of MAX and MXene was analyzed using X-ray diffraction (XRD, Ultima IV, Rigaku, Tokyo, Japan) with X-rays of Cu Kα at 0.1542 nm and O1s X-ray photoelectron (XPS, AXIS Supra, Kratos, Manchester, UK) with X-ray source of Al Kα (1486.71 ev). Fourier transform infrared spectroscopy with mode of attenuated total reflection (ATR-FTIR, Nicolet 6700, Thermo Scientific, Waltham, MA, USA) was used to demonstrate the formation of coordination bonds.

### 2.5. Mechanical Measurements

The mechanical properties of the hydrogels were all tested by a tensile tester (5567, Instron, Norwood, MA, USA). The hydrogel samples were shaped in the form of dumbbells and were subjected to tensile tests at a temperature of 25 °C at a speed of 100 mm/min. Young’s modulus is the initial slope value of the stress–strain curve. Uniaxial tensile tests were carried out at different tensile rates to investigate the dynamic nature of hydrogels. For cyclic tensile tests, the hydrogels were stretched to a given length and then returned to the original position at the same tensile rate. The total energy is the integral area of the curve at the maximum strain with respect to the *X*-axis; the dissipated energy is the integral area of the hysteresis loop in one cycle; the ratio of the dissipated energy to the total energy is defined as the energy dissipation ratio. For the recovery properties test, the samples were stored in an environmental chamber for a period of time to rest without any disturbance before the next loading procedure.

### 2.6. Rheological Measurements

Rheological tests were performed by a rotational rheometer (MCR302, Anton Paar, Graz, Austria). Frequency sweeps in a range of 0.1–100 Hz were performed at a fixed strain of 0.1% and a fixed temperature of 25 °C. The spectra of storage modulus (G′) and loss modulus (G″) were recorded, where the loss factor (tanδ) is the ratio of G″ to G″. Then, 0.1% and 100% step-shear strain sweeps were performed on the hydrogel samples at a temperature of 25 °C and at a frequency of 1 Hz to investigate the self-recovery ability.

## 3. Results and Discussion

Prior to the preparation of hydrogels, we prepared Ti_3_C_2_Tx MXene nanosheets by selective etching: the aluminum layer in the parent Ti_3_AlC_2_ powder was shaved off, ultrasonically processed, and peeled into ultrathin MXene nanosheets. As shown in Figure 1a, when a red laser beam was illuminated onto a 0.5 wt% mass fraction of MXene dispersion, it exhibited a pronounced Tyndall effect, as evidenced by the nanoscale dimension. In the XRD pattern, the characteristic peaks of the raw material Ti_3_AlC_2_ disappeared near 2θ = 39°, while the characteristic peaks at 2θ = 10° were significantly shifted to a lower angle, and a series of new peaks appeared near 2θ = 6.9° (Figure 1b). The O1s region in the XPS spectrum of the MXene is fitted by components corresponding to C-Ti-(OH)_x_ at BE = 532.1 eV, C-Ti-O_x_ at BE = 531.2 eV, and TiO_2_ at BE = 529.9 eV (Figure 1c). The above results effectively confirmed that MAX was successfully exfoliated into a single layer of MXene nanosheets, laying a foundation for the development of high-performance hydrogels enabled by MXene nanosheets.

Tough and robust metallosupramolecular hydrogels resulting from MXene were fabricated by a two-step process (Figure 2). In the presence of MXene nanosheets, an aqueous solution of AM and AA was subjected to in situ free-radical polymerization at room temperature using KPS as an initiator and TMEDA as a promoter. The pristine hydrogels were then incubated in 0.1 mol/L FeCl_3_ solution to form a physical cross-linked network, and then immersed in excess deionized water to remove ions and reactants not involved in coordination and to reach equilibrium. The eventually generated hydrogels were noted as M-*f*-AA-Fe^3+^ gels, where *f* represents the mass percent of feeding MXene. A large number of polar groups, such as COOH, OH, etc., are present on the surface of MXene nanosheets, which can both form hydrogen-bonding interactions with the hydrogel polymer backbone and cross-link the hydrogel with MXene via Fe^3+^ ions. Therefore, the coordination interactions in the hydrogel network exist not only between the polymer chains, but also between the polymer chains and the MXene nanosheets, which synergistically strengthen and toughen the hydrogel.

The formation of supramolecular networks in the above metallosupramolecular hydrogels is attributed to strong coordination interactions between carboxyl groups and Fe^3+^ ions. This can be demonstrated by the Fourier transform infrared spectroscopy (FTIR) spectra (Figure 3). On the FTIR spectra of the uncoordinated M-0.4-AA hydrogel, the peaks at 1424 cm^−1^, 1456 cm^−1^, 1615 cm^−1^, and 1662 cm^−1^ are assigned to the symmetric and asymmetric stretching vibrations of the carboxyl group, respectively. After introducing Fe^3+^, these peaks shifted to 1418 cm^−1^, 1452 cm^−1^, 1610 cm^−1^, and 1655 cm^−1^, respectively, showing the formation of coordination interactions between Fe^3+^ and COOH in the hydrogel network. The shift of the infrared characteristic peaks effectively demonstrates that the introduction of iron ions can effectively construct metal-coordination complexes.

The mechanical properties of these tough and robust metallosupramolecular hydrogels can be tuned over a wide range by adjusting the feeding mass fraction of MXene. The total concentration of all the monomers was fixed at 4.5 mol/L. As shown in Figure 4a–d, the mechanical properties of all these hydrogels are superior compared to similar hydrogels. With the increase in the mass fraction *f* of MXene from 0.2% to 0.6%, the fracture strength of the hydrogel increased from 4.18 MPa to 5.87 MPa, the Young’s modulus increased from 1.78 MPa to 3.89 MPa, while the fracture strain gradually decreased from 508% to 353%. It is worth noting that the mechanical properties of the hydrogels decreased when *f* exceeded 0.6%. When the MXene content is too high, the carboxylate content in the hydrogel network increases dramatically, and the coordination cross-linking between carboxylate and Fe^3+^ makes the hydrogel significantly denser and more brittle. In short, these tough and robust metallosupramolecular hydrogels can be easily prepared with a wide range of tailored mechanical properties via changing the feeding mass fraction of MXene, which may facilitate their practical application in load-bearing fields.

The excellent mechanical properties are attributed to the presence of a superior energy dissipation mechanism in the hydrogel network. Cyclic tensile experiments at different strains (50%, 100%, 150%, 200%, 250%, 300%) were carried out. As illustrated in Figure 5a, the hydrogel demonstrates an excellent energy dissipation capacity at both large and small strains with a large hysteresis loop, which is attributed to the contribution of the ferric ion-carboxylate coordination network. As the fixed strain increased from 50% to 300%, the hydrogel dissipated more and more energy in a single cycle of stretching, i.e., more and more catboxylate-Fe^3+^ coordination complexes were broken (Figure 5b). Notably, the energy dissipation ratio also increased from 40% to 80% when the tensile strain increased from 50% to 300% (Figure 5c). A six-fold increase in tensile strain resulted in a near-doubling of the energy dissipation ratio, demonstrating the strong energy dissipation capacity in the hydrogel network after the introduction of MXene and coordination interactions. Moreover, the hydrogel showed outstanding fatigue resistance. Therefore, ten consecutive loading–unloading tests were carried out on the prepared specimens to investigate their fatigue resistance. As shown in Figure 5d, the hysteresis loops of the prepared hydrogels decreased significantly in the second and subsequent cycles. This trend indicates that a significant dissociation of the dynamic physical cross-linking occurred during the first loading–unloading process, and this dynamic cross-linking could not be immediately recovered from the damage without resting due to the viscoelasticity of the polymer system. That is to say, when resting for a period of time, we speculate that this dissociated dynamic coordination complex can be reattached, thus conferring certain recovery properties to the hydrogel, as we will discuss later. However, after 10 consecutive loading–unloading tensile tests at large strains, the hydrogels still did not show an obvious crack extension and fracture, and could still reach the neighborhood of the maximum initial stress at the maximum strain, which demonstrate that the hydrogels could withstand the persistent load-bearing test. The above results are sufficient to show that the hydrogel has satisfactory and excellent fatigue resistance, which lays a foundation for expanding the wide range of applications of the metal-coordinated hydrogels.

Due to the dynamic nature of coordination bonds, the mechanical properties of metallosupramolecular hydrogels show clear dependence on the rate of tensile stretch. As shown in Figure 6, the hydrogels’ tensile strength and Young’s modulus increased from 3.97 MPa and 1.65 MPa to 6.27 MPa and 3.96 MPa, respectively, while the fracture strain decreased from 464% to 320% as the stretch rate increased from 5 mm/min to 500 mm/min. At low rates, the hydrogels are soft and elastic, while at high rates, they become hard and brittle. This rate-dependent behavior is due to the dynamic time scale of the coordination bonds in relation to the tensile stretch rate. At lower tensile stretch rates, the coordination interactions have enough time to dissociate and reorganize when damage occurs, thus continuously dissipating energy and dispersing stress concentrations, resulting in the hydrogels exhibiting superior ductility, i.e., larger elongation at break; at higher stretching rates, the coordination interactions remain intact, and there is almost no time for dissociation to be completed when damage occurs, resulting in a permanent, physical cross-linking effect that resists deformation, leading to a significant increase in modulus and stress, and a decrease in elongation at break.

As described in the previous section, these metallosupramolecular hydrogels are tough, robust, and exhibit excellent fatigue resistance and dynamic properties, which contribute to their self-recovery capability. We performed cyclic stretching tests on the hydrogels at different resting times to examine their recovery properties. As illustrated in Figure 7a, it can be observed that the hydrogel shows minimal energy dissipation during the second consecutive stretching process, and the loading curves and unloading curves almost overlap, which shows an obvious low hysteresis. Notably, after 120 min of rest without external interference, the hydrogel’s loading–unloading hysteresis loop almost perfectly overlaps with the original pathway, indicating its remarkable self-recovery ability. Thereby, when the hydrogel is damaged and the dynamic metal coordination bonds are broken, it does not recover immediately but takes some time to reorganize.

In addition, to further investigate the recovery properties, we also conducted continuous step shear measurements at different shear strains, as shown in Figure 7b. When the oscillatory strain increases from 0.1% to 100%, the hydrogel transitions from a gel to a sol state with a sharp decrease in modulus. Conversely, when the oscillatory strain decreases from 100% to 0.1%, the hydrogel returns from a sol state to the gel state, and its storage modulus (G′) and loss modulus (G″) rapidly recover to their original values. Repeating this cycle of small and large shear strains three times, the hydrogel can always recover quickly, further verifying the excellent self-recovering behavior of the hydrogel samples. This behavior also demonstrates the dynamic “dissociation-restructuring” nature of the coordination bonds during deformation. Interestingly, the modulus of the hydrogels slightly increased after recovery from one single shear damage, which was attributed to the process of coordination bond dissociation and reorganization that optimized the coordination network, with more triple-dentate coordination formed between Fe^3+^ and carboxylic acid. The above results further highlight the hydrogel’s excellent self-recovery ability. Additionally, these tough and robust metallosupramolecular hydrogels exhibit exceptional viscoelasticity. A constant-temperature sweep frequency test was performed, as illustrated in Figure 7c. For this tough and robust metallosupramolecular hydrogel, its storage modulus (G′) is consistently larger than the loss modulus (G″) over the frequency range, indicating its solid nature. As the frequency decreases, the G′ of the hydrogel gradually decreases while the G” gradually increases, which demonstrates that the hydrogel has obvious frequency dependence and exhibits good viscoelastic behavior.

Figure 7d illustrates that the value of the loss factor, i.e., Tan δ (calculated by G″/G′), decreases with increasing frequency and finally reaches a plateau. In the low-frequency region, the coordination bonds can have enough time to experience “dissociate-reorganize” repeatedly, so they can dissipate the energy effectively, and the value of Tan δ is relatively large; while as the frequency increases, the dissociation–reorganization process of the coordination bonds is not enough to dissipate the energy effectively, and the value of Tan δ gradually decreases. When the critical threshold is reached, the Tan δ value tends to reach a plateau. Therefore, the energy dissipation capacity of hydrogels decreases with increasing frequency, which is consistent with the results of tensile experimental tests.

## 4. Conclusions

In conclusion, we synthesized a tough and robust metallosupramolecular hydrogel by a two-step method. The premier hydrogels were fabricated by in situ radical polymerization of AM and AA at room temperature. Thereafter, the two-dimensional Ti_3_C_2_T_x_ MXene nanosheets, as multifunctional cross-linking agents, were combined with Fe^3+^ to promote the formation of the cross-linking network: Fe^3+^-mediated coordination cross-links between the MXene nanosheets and poly(AM-co-AA) chains and carboxylate-Fe^3+^ cross-links between the polymer backbone chains. The MXene nanosheets act as dense cross-linking centers to gradually dissipate mechanical energy and redistribute stress concentration, i.e., transfer the applied stresses to more polymer chains. This ingenious design principle reported in this work gives the obtained metallosupramolecular hydrogels outstanding mechanical properties: tensile strength of up to 5.87 MPa, strain at break of up to 508%, excellent self-recovery ability, and good viscoelasticity. Such tough and robust mechanical properties will effectively broaden those of metallosupramolecular hydrogels’ application scenarios.

## Figures and Tables

**Figure 1 polymers-15-04025-f001:**
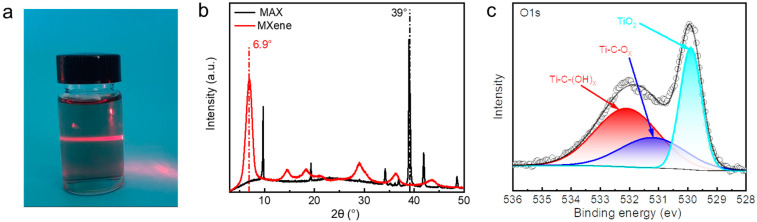
(**a**) Tyndall effect of MXene nanosheets suspension (0.5 wt%). (**b**) XRD spectra of Ti_3_AlC_2_ MAX phase and Ti_3_C_2_T_x_ MXene nanosheets. (**c**) XPS spectra of Ti_3_C_2_T_x_ MXene nanosheets.

**Figure 2 polymers-15-04025-f002:**
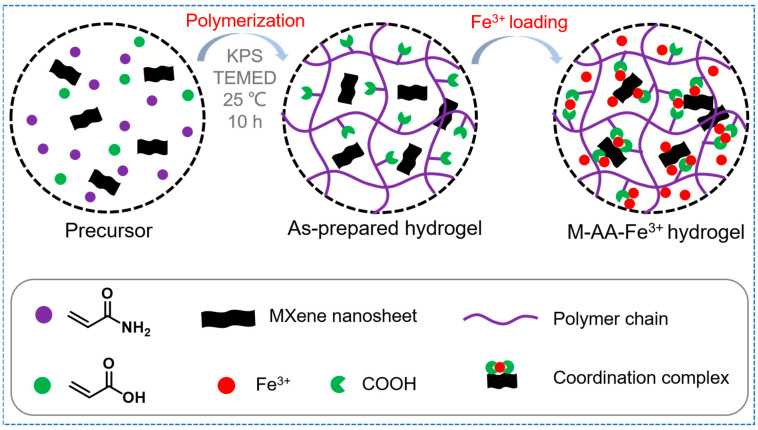
Schematic illustration for the fabrication of PAI-Fe^3+^ hydrogels.

**Figure 3 polymers-15-04025-f003:**
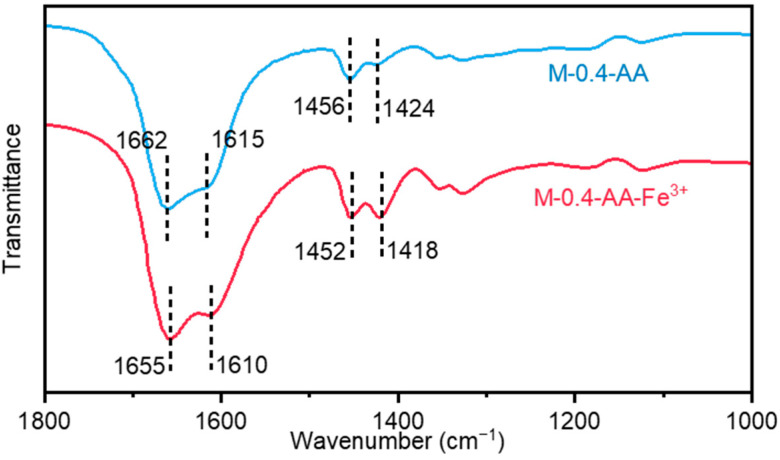
FTIR spectra of the uncoordinated (M-0.4-AA) and coordinated (M-0.4-AA-Fe^3+^) hydrogels.

**Figure 4 polymers-15-04025-f004:**
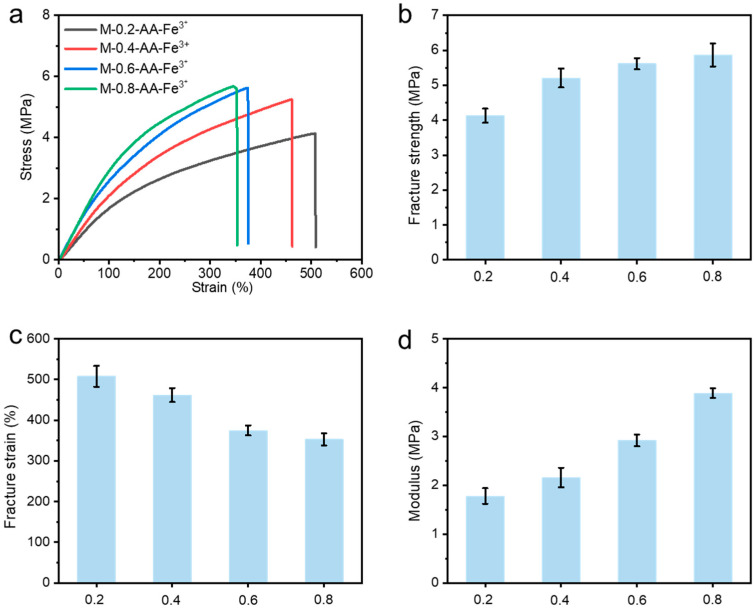
(**a**) Tensile stress–strain curves and corresponding (**b**) fracture strength, (**c**) fracture strain, and (**d**) modulus parameters of the M-*f*-AA-Fe^3+^ hydrogels with different mass ratios of MXene.

**Figure 5 polymers-15-04025-f005:**
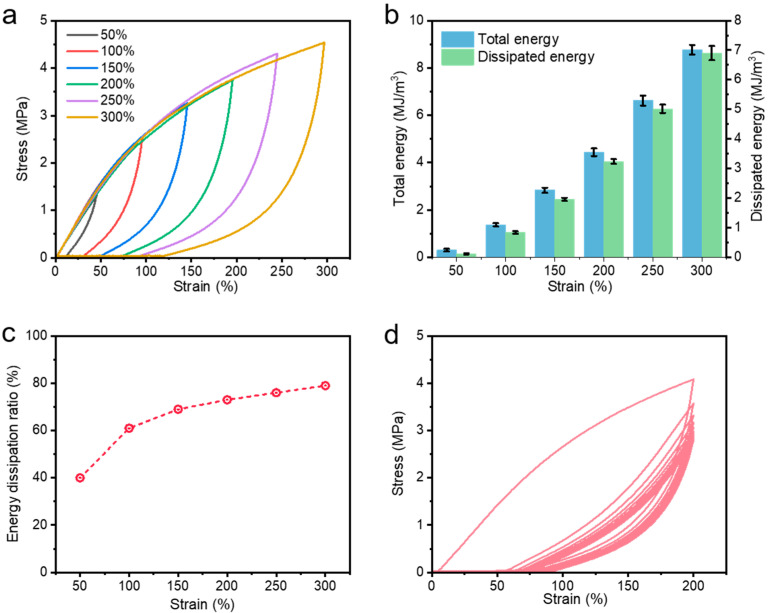
(**a**) Cyclic tensile loading–unloading curves under different strains on M-4.0-AA-Fe^3+^ gels, (**b**) corresponding total energy, dissipated energy, and (**c**) energy dissipation ratio. (**d**) Fatigue resistance by ten successive loading–unloading cycles of the M-4.0-AA-Fe^3+^ samples.

**Figure 6 polymers-15-04025-f006:**
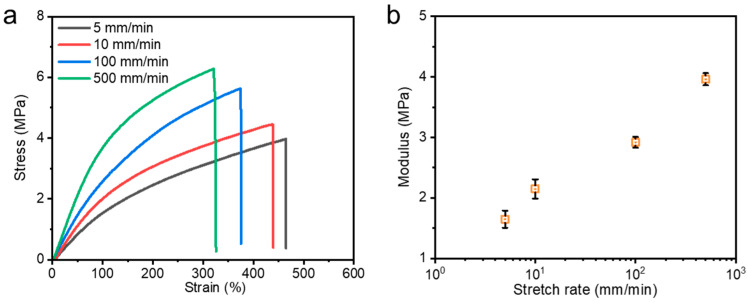
(**a**) Nominal stress–strain curves of the M-0.4-AA-Fe^3+^ hydrogel under different initial stretch rates ranging from 5 mm/min to 500 mm/min and (**b**) corresponding modulus as a function of initial strain rate.

**Figure 7 polymers-15-04025-f007:**
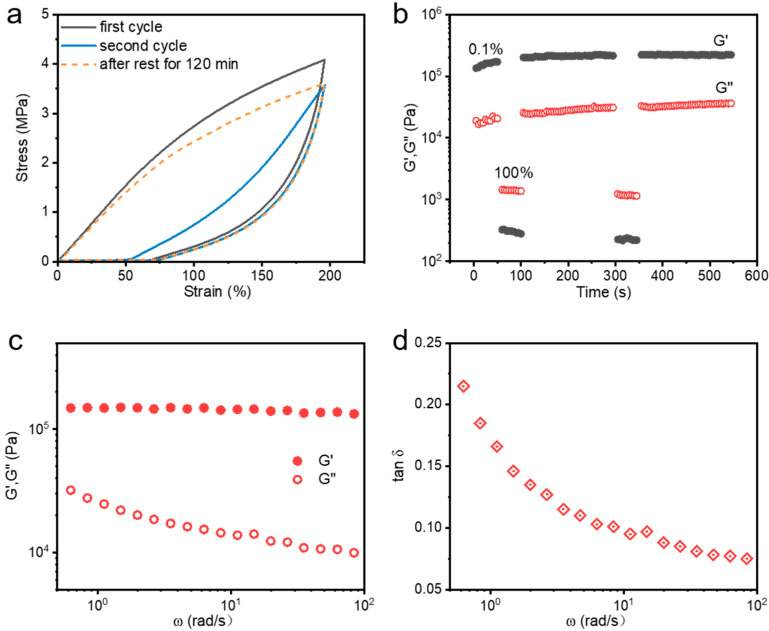
(**a**) Cyclic tensile loading–unloading curves at 200% strain with different resting times between two continuous tests. (**b**) G′ and G″ dependence on time in continuous step shear strain test for M-0.4-AA-Fe^3+^ hydrogel with repeated small oscillation strain (strain = 0.1%, frequency = 1.0 Hz) and large oscillation strain (strain = 100%, frequency = 1.0 Hz). (**c**) Viscoelastic moduli and (**d**) loss factor as a function of the strain frequency for the M-0.4-AA-Fe^3+^ hydrogel.

## Data Availability

The research data can be obtained by Biqiang Jin and Wenqiang Wu.

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
