# Peer review of "Tough and Robust Metallosupramolecular Hydrogels Enabled by Ti3C2Tx MXene Nanosheets"

_polymers, 2023, doi:10.3390/polym15194025_

Round 1

Reviewer 1 Report

The paper “Tough and Robust Metallosupramolecular Hydrogels Enabled by Ti3C2Tx MXene Nanosheets” by Biqiang Jin, Wenqiang Wu, Zhaoyang Yuan and Changcheng Wang presents an experimental study on the mechanical properties of polyacrylamide hydrogels with embedded inorganic nanosheets. Stress-strain dependencies,  mechanical hysteresis loops, and dynamic moduli at oscillatory deformation are presented and discussed in their dependence on the load of inorganic filler. In principle the paper fits the scope of the journal. The data are new and might be interesting for the applications of hydrogels in bioengineering purposes.

 However, the paper is unacceptable for publication in its present state. Extensive major revision should be provided.

 The major drawback is that the gel materials under study are not properly and sufficiently characterized.

 1. Inorganic compound  Ti3C2Tx MXene is not among common widely known chemical substances. There is a couple of references concerning it but they also do not help much in understanding the properties of that material. It looks like it is a derivative of titanium carbide. Chemical composition should be clarified. It is stated that Ti3C2Tx MXene is used in form of nanosheets but no usual confirmation of that is provided. Typically, SEM or TEM micrographs are given. Specific surface area might also help. XRD pattern is given but erroneously named “spectrum”. Figure 1 is a diffractogram and it is not properly discussed. What is the type of crystalline lattice of MAX and MXene? How does it changes in preparation of nanosheets? The strongest peak at 6.9 deg 2Theta corresponds to very large crystal period - around 1.3 nm by Bragg’s equation using first order diffraction index. What does it correspond to? MXene is stated as a chemically modified but no characterization of chemical modification is provided.

 2. In preparation of hydrogel no total volume of solution is given and it is not clear to what does the fraction of MXene relate. Proper characterization of hydrogels is missing. Their fundamental property is the swelling ratio or in other words water uptake. It is an indicator of the networking density of hydrogel. It is not shown how did the networking density change during the steps of preparation procedure. Such change might substantially influence the mechanical properties of gels.

 3. There are uncertainties in the sketch in Figure 3. Acrylic acid comonomers should be incorporated in sub-chains of the AM network but not attached to them. It is shown that MXene nanosheets are smaller than the mesh-size of the network but no proof for that conclusion is given. An estimation for the size of nanosheets (i.e. TEM) and for the mesh-size (i.e. using Flory-Rhenner equation) might be appropriate.

 Technical remark: Lines 158 - 164 are the exact repetition of lines 144 - 150.

 4. FTIR spectrum (Figure 3) for an unclear reason does not reveal a very strong peak at 1790 cm(-1) which is characteristic for C=O group in amides. The peaks at 1650 and 1450 cm(-1) are usually related to the ionized -COO(-1) carboxylate residue. The question is then whether AA comonomer was initially neutralized in the synthesis? It was not mentioned in the experimental section. If it was done then hydrogel was a polyelectrolyte one and it should substantially shrink in the presence of multivalent Fe(+3) ions. This was not marked out either.

 Presentation of mechanical data might also be improved

 5. It is stated in Line 189 that the mechanical properties decrease if the content of MXene is above 0.6% It is not supported by Figure 5a which shows that fracture stress gradually increases and fracture strain decreases over the entire range.Also, in this respect a blank gel with no MXene filler seems appropriate as a reference sample. Right and left Y-axes in Figure 5b are the same but with different scale which is confusing. Better present energy dissipation ratio rather than dissipation energy. What is G' and what is G" in Figure 7b? Y-Axis title  incorrect in figure 7d.

 6. Mechanical test revealed that deformation of these gels is accompanied with huge energy losses due to dissipation in hysteresis. Figure 7a shows that a 2-hour period is needed for the elastic recovery. Why these mechanical features are so “excellent” or “outstanding” as stated. If it were for some certain application then it better be clarified.

Author Response

The paper “Tough and Robust Metallosupramolecular Hydrogels Enabled by Ti3C2Tx MXene Nanosheets” by Biqiang Jin, Wenqiang Wu, Zhaoyang Yuan and Changcheng Wang presents an experimental study on the mechanical properties of polyacrylamide hydrogels with embedded inorganic nanosheets. Stress-strain dependencies,  mechanical hysteresis loops, and dynamic moduli at oscillatory deformation are presented and discussed in their dependence on the load of inorganic filler. In principle the paper fits the scope of the journal. The data are new and might be interesting for the applications of hydrogels in bioengineering purposes.

 However, the paper is unacceptable for publication in its present state. Extensive major revision should be provided. The major drawback is that the gel materials under study are not properly and sufficiently characterized.

Authors’ response: We appreciate the very helpful comments of Reviewer 1. We have made some explanations and revised the manuscript according to the reviewer’s comments. All the changes are marked in the text and a detailed list to answer each of the questions is provided below.

  1. Inorganic compound  “Ti3C2Tx MXene” is not among common widely known chemical substances. There is a couple of references concerning it but they also do not help much in understanding the properties of that material. It looks like it is a derivative of titanium carbide. Chemical composition should be clarified. It is stated that Ti3C2Tx MXene is used in form of nanosheets but no usual confirmation of that is provided. Typically, SEM or TEM micrographs are given. Specific surface area might also help. XRD pattern is given but erroneously named “spectrum”. Figure 1 is a diffractogram and it is not properly discussed. What is the type of crystalline lattice of MAX and MXene? How does it changes in preparation of nanosheets? The strongest peak at 6.9 deg 2Theta corresponds to very large crystal period - around 1.3 nm by Bragg’s equation using first order diffraction index. What does it correspond to? MXene is stated as a chemically modified but no characterization of chemical modification is provided.

Authors’ response: Indeed, MXene and graphene oxide, is a nanofiller containing a large number of polar groups. It has been reported in many papers and has been used to synthesize various types of materials. XRD test was performed to demonstrate the success in etching and exfoliation of MAX into monolayers of MXene. The different mass fractions of MXene were the feeding amounts to compound them in a hydrogel. To further demonstrate the successful preparation of monolayer MXene, we supplemented the X-ray photoelectron spectroscopy (XPS) test and a demonstration of the Tyndall effect of the Mxene dispersion. The details are as follows:

We have added related tests and relative descriptions in manuscript: As shown in Figure 1a, the MXene dispersions exhibit a pronounced Tyndall effect, as evidenced by their nanoscale dimension. In the XRD spectrum, the characteristic peaks of the raw material Ti3AlC2 disappeared near 2θ=39°, while the characteristic peaks at 2θ=10° were shifted to a lower angle significantly, and a series of new peaks appeared near 2θ=6.9° (Figure 1b). The O1s region in the XPS spectrum of the MXene is fitted by components corresponding to C-Ti-(OH)x at BE = 532.1 eV, C-Ti-Ox at BE = 531.2 eV, and TiO2 at BE = 529.9 eV (Figure 1c).

Figure 1. (a) Tyndall effect of MXene nanosheeets suspension (0.5 wt%). (b) XRD spectra of Ti3AlC2 MAX phase and Ti3C2Tx MXene nanosheets. (c) XPS spectra of Ti3C2Tx MXene nanosheets.

  1. In preparation of hydrogel no total volume of solution is given and it is not clear to what does the fraction of MXene relate. Proper characterization of hydrogels is missing. Their fundamental property is the swelling ratio or in other words water uptake. It is an indicator of the networking density of hydrogel. It is not shown how did the networking density change during the steps of preparation procedure. Such change might substantially influence the mechanical properties of gels.

Authors’ response: We are sorry for omitting the concentration of the precursor solution used to prepare the hydrogel. The concentration of precursor liquid was strictly controlled to 4.5 mol/L for all samples. Moreover, the concentration of MXene, i.e., f, is the feeding mass fraction to the total monomer. In this manuscript, we concentrated on the mechanical properties and viscoelasticity of the MAX once it was successfully etched and exfoliated into a monolayer of MXene. The network crosslink density can be reflected as the Young's modulus calculated from the stress-strain curves.

  1. There are uncertainties in the sketch in Figure 3. Acrylic acid comonomers should be incorporated in sub-chains of the AM network but not attached to them. It is shown that MXene nanosheets are smaller than the mesh-size of the network but no proof for that conclusion is given. An estimation for the size of nanosheets (i.e. TEM) and for the mesh-size (i.e. using Flory-Rhenner equation) might be appropriate. Technical remark: Lines 158 - 164 are the exact repetition of lines 144 - 150.

Authors’ response: Thank you for pointing this out! We are sorry for including uncertainties in Figure 3. We have replaced the circular representation of the carboxylic acid group with other graphics to more visually represent its involvement in coordination with the iron ion. In addition, the size of the MXene nanosheets and hydrogel network is not a clear-cut judgment; the purple lines to represent the polymer chains, and the black squares to represent MXene, the network actually has a lot of intertwined polymer chains. We just draw it this way in order to visualize the structure of the hydrogel network. The relationship between the nanofiller and the hydrogel pore size can be investigated later. The details are as follows:

Figure 2. Schematic illustration for the fabrication of PAI-Fe3+ hydrogels.

  1. FTIR spectrum (Figure 3) for an unclear reason does not reveal a very strong peak at 1790 cm(-1) which is characteristic for C=O group in amides. The peaks at 1650 and 1450 cm(-1) are usually related to the ionized -COO(-1) carboxylate residue. The question is then whether AA comonomer was initially neutralized in the synthesis? It was not mentioned in the experimental section. If it was done then hydrogel was a polyelectrolyte one and it should substantially shrink in the presence of multivalent Fe(+3) ions. This was not marked out either.  Presentation of mechanical data might also be improved

Authors’ response: Thank you for pointing this out! The C=O stretching vibrational peaks appear in a specific range, and this thesis, like other reported methods, only demonstrates that the carboxylate group produces an interaction, i.e., a coordination interaction, by the shift of the characteristic peaks. The carboxylate was not neutralized during polymerization. Moreover, we have also optimized the writing of the full text.

  1. It is stated in Line 189 that the mechanical properties decrease if the content of MXene is above 0.6% It is not supported by Figure 5a which shows that fracture stress gradually increases and fracture strain decreases over the entire range.Also, in this respect a blank gel with no MXene filler seems appropriate as a reference sample. Right and left Y-axes in Figure 5b are the same but with different scale which is confusing. Better present energy dissipation ratio rather than dissipation energy. What is G' and what is G" in Figure 7b? Y-Axis title  incorrect in figure 7d.

Authors’ response: Thank you for your good comments! Indeed, the left Y-axis represents the total energy dissipated at the corresponding strain, being the area of integration of the curve with the X-axis at the maximum strain; the right Y-axis one represents the area enclosed by the hysteresis circle at that strain. The ratio of the two is the energy dissipation ratio, which is reflected in Fig. 5c. Moreover, we have modified the Y-Axis title of Fig. 7b and 7d. We have modified the figures as follow:

Figure 5. (a) Cyclic tensile loading-unloading curves under different strains on M-4.0-AA-Fe3+ gels, (b) corresponding total energy, dissipated energy and (c) energy dissipation ratio. (d) Fatigue resistance by ten successive loading-unloading cycles of the M-4.0-AA-Fe3+ samples.

Figure 7. (a) Cyclic tensile loading–unloading curves at 200 % strain with different resting times between two successive measurements. (b) G′ and G′′ dependence on time in continuous step shear strain measurements for M-0.4-AA-Fe3+ hydrogel with repeated small oscillation strain (strain = 0.1 %, frequency = 1.0 Hz) and large oscillation strain (strain = 50 %, frequency = 1.0 Hz). (c) Viscoelastic moduli (G′, solid; G′′, open) and (d) loss factor (tan δ = G′′/G’) as a function of the strain frequency for the M-0.4-AA-Fe3+ hydrogel.

  1. Mechanical test revealed that deformation of these gels is accompanied with huge energy losses due to dissipation in hysteresis. Figure 7a shows that a 2-hour period is needed for the elastic recovery. Why these mechanical features are so “excellent” or “outstanding” as stated. If it were for some certain application then it better be clarified.

Authors’ response: Thank you for your good comments! Fe3+-carboxylate is a highly dynamic noncovalent bond, which can be reflected in the rate sensitivity of hydrogels, i.e. Figure 6. When they are damaged, they can generally recover their original mechanical properties after a period of rest.

Reviewer 2 Report

This manuscript developed ‘high’ strength metallosupramolecular hydrogels. The study investigated the chemical structures, mechanical properties and rheology characteristics of the hydrogels.

some comments are listed here:

1.       The manuscript is well-organized and easy to read.

2.       In the introduction section, some previous studies were listed which showed that the previous developed hydrogels have higher strength than the proposed study. It’s suggested that the motivation and innovation of the study should be discussed in the manuscript.

the mauscript is easy to read

Author Response

This manuscript developed ‘high’ strength metallosupramolecular hydrogels. The study investigated the chemical structures, mechanical properties and rheology characteristics of the hydrogels.

some comments are listed here:

  1. The manuscript is well-organized and easy to read.
  2. In the introduction section, some previous studies were listed which showed that the previous developed hydrogels have higher strength than the proposed study. It’s suggested that the motivation and innovation of the study should be discussed in the manuscript

Authors’ response: We appreciate the very helpful comments of Reviewer 2. We have added the motivation and innovation discussion in Introduction.

Reviewer 3 Report

Dear Author

your article cann't be accepted in the present form please follow the following comments

- palagrism chek for your article is about 34 %, it is no accepted please reconsider your article and check the plagrism again before resubmitted , as the first palgirismed article (your own article ) is about 8 %

-in the synthesis of hydrogel , what is used volume of Fe ion

- how you got the final hydrogel including Maxin , please write the process and condition including the used amount to help other researchers

- What is the difference between this new sheets your former sheet and other sheets published before

Author Response

Dear Author

your article cann't be accepted in the present form please follow the following comments

- palagrism chek for your article is about 34 %, it is no accepted please reconsider your article and check the plagrism again before resubmitted, as the first palgirismed article (your own article) is about 8 %

-in the synthesis of hydrogel, what is used volume of Fe ion

- how you got the final hydrogel including Maxin, please write the process and condition including the used amount to help other researchers

- What is the difference between this new sheets your former sheet and other sheets published before

Authors’ response: We appreciate the very helpful comments of Reviewer 3. We are sorry for the palagrism check is 34% and we have modified the manuscript. The concentration of the FeCl3 solution is 0.1 mol/L and the volume is not a certain value. The properties of the hydrogel are not related to the volume of iron ions; equilibrium is sufficient. We have added detail descriptions about hydrogel fabrication.

Round 2

Reviewer 1 Report

The authors have provided acceptable response to my comments and have improved the paper. It might be published.

Reviewer 3 Report

Dear Author

Authors have did all requested comments, so the article can be accepted for publication